# An In Vitro and In Silico Investigation about *Monteverdia ilicifolia* Activity against *Helicobacter pylori*

**DOI:** 10.3390/antibiotics12010046

**Published:** 2022-12-28

**Authors:** Mariana Nascimento de Paula, Taísa Dalla Valle Rörig Ribeiro, Raquel Isolani, Daniela Cristina de Medeiros Araújo, Augusto Santos Borges, Gisele Strieder Philippsen, Rita de Cássia Ribeiro Gonçalves, Rodrigo Rezende Kitagawa, Flavio Augusto Vicente Seixas, João Carlos Palazzo de Mello

**Affiliations:** 1Laboratory of Pharmaceutical Biology, Department of Pharmacy, State University of Maringá, Maringá 87020-900, Brazil; 2Ingá University Center, Maringá 87035-510, Brazil; 3Department of Pharmaceutical Science, Federal University of Espirito Santo, Vitória 29047-105, Brazil; 4Federal University of Paraná, Jandaia do Sul 86900-000, Brazil; 5Department of Technology, State University of Maringá, Umuarama 87501-390, Brazil

**Keywords:** *Monteverdia ilicifolia*, Maytenus ilicifolia, antioxidant capacity, condensed tannins, glycosylated flavonoids, bioinformatics

## Abstract

*Monteverdia ilicifolia* is a Brazilian native plant, traditionally used to treat gastric diseases that are now associated with *Helicobacter pylori* and are commonly associated with several human diseases. We point out the *M. ilicifolia* extract as active against *H. pylori*. The crude extract produced with acetone:water presented the best *H. pylori* inhibitory activity of all five extracts (MIC 64 µg/mL). The ethyl-acetate fractions from crude extracts produced with ethanol and acetone showed a MIC of 64 µg/mL. Both ethyl-acetate fractions and the crude extract produced with acetone showed an antioxidant capacity of between 14.51 and 19.48 µg/mL in the DPPH assay. In the FRAP assay, two ethyl-acetate fractions (EAF2 and EAF4) presented the antioxidant capacity of 5.40 and 5.15 mM Trolox/g of extract. According to the results obtained from the antioxidant and antibacterial assays, two fractions (EAF2 and nBF5) were analyzed by mass spectrometry and confirmed the presence of monomeric, dimeric, trimeric tannins, and glycosylated flavonoids. Some compounds were tested using bioinformatics to evaluate the best enzyme inhibitors and the molecular interaction between the enzyme and the tested ligands. The presence of these polyphenol compounds could play an important role in antioxidant and inhibitory capacities against *H. pylori* and can be used to assist in the treatment or prevention of infection by *H. pylori.*

## 1. Introduction

*Helicobacter pylori* is a Gram-negative bacillus that can contribute to the development of human diseases, such as gastric ulcers, gastritis, gastric adenocarcinoma, and lymphoma. The bacteria express some virulence factors that expand the possibilities of the bacteria interacting with the host cells. One of those factors allows the bacteria to neutralize the acidity of gastric secretion, hydrolyzing urea to ammonium and CO_2_, promoting the adjustment of pH in the stomach to neutral, and producing a cytopathic effect in the stomach cells. This mechanism allows the bacteria to survive in stomach conditions [1,2].

In 2015 approximately 50% of the world’s population (4.4 billion) was estimated to be infected with *H. pylori*. In developing and newly industrialized countries, the prevalence is higher than in developed countries. This difference reflects the level of urbanization, sanitation, access to clean water, and varied socioeconomic statuses. Brazil has one of the highest infection prevalences in Latin America and the Caribbean (71.2%) [3].

The first-line treatment for *H. pylori* infection is one proton pump inhibitor, amoxicillin, and clarithromycin for 14 days. As an alternative, treatment can be used in quadruple therapy with bismuth for 10–14 days and concomitant therapy for 14 days [4]. Several therapeutic strategies have been proposed to increase bacteria eradication rates. *H. pylori* infection is a unique therapeutic challenge. The infection eradication failure rate remains as high as 5–20%, along with frequent relapses in gastric ulcers even after the discerned complete healing. Natural products have great potential to serve as alternative sources of bioactive compounds and can be used to treat infectious diseases. The rapid growth of bacterial resistance to antimicrobials is occurring worldwide, putting the effectiveness of these drugs at risk, and making the treatment of bacterial infections a major challenge [5].

Time and effort have been spent to find effective alternative treatments against the bacterium and to intervene in the initial infection phase, which is becoming very popular in the search for alternative therapies, including medicinal plants that are active against the bacterium, to improve the effectiveness of bacteria eradication [6].

*Monteverdia ilicifolia* (Mart. ex Reissek) Biral, previously known as *Maytenus ilicifolia* Mart. ex Reissek Celastraceae is an Atlantic Forest native plant. In Brazil it is popularly known as espinheira-santa, cancerosa, and cancorosa-de-sete-espinhos [7,8]. *M. ilicifolia* is popularly used due to its analgesic, antiulcer, antitumor, aphrodisiac, antispasmodic, contraceptive, antiulcerogenic, diuretic, and healing properties. In 1988 the medicinal properties of *M. ilicifolia* were proven to treat gastrointestinal diseases, especially gastritis and ulcers, and in 2009 the plant was included in the National List of Medicinal Plants of Interest to the Unified Health System, associated with the Health Ministry, in Brazil [9].

*M. ilicifolia* ethnopharmacological information is well documented, with pharmacological, microbiological, and phytochemistry studies. The leaves of *M. ilicifolia* are rich in triterpenes, chromones, alkaloids, essential oils, and polyphenols, as well as tannins and flavonoids. The identified compounds in the vegetal specie have shown antiulcerogenic, antioxidant, gastroprotection, antifungal, anticarcinogenic, anti-leishmanicidal, trypanomicide, and anti-inflammatory properties [10,11,12,13,14,15,16].

Polyphenol extracts from medicinal plants have been studied for their antimicrobial activity. The results have been positive, showing their ability to inhibit the growth of *Vibrio cholerae*, *Streptococcus mutans*, *Campylobacter jejuni*, *Escherichia coli*, *Candida albicans*, *Staphylococcus aureus*, *Porphyromonas gingivalis*, and *Helicobacter pylori*, among others [17,18].

This study aimed to evaluate the antibacterial (*H. pylori*) and antioxidant activity, as well as identify compounds from *M. ilicifolia* extracts and fractions.

## 2. Results and Discussion

### 2.1. Epicatechin Determination Using HPLC

According to the Brazilian Pharmacopeia [19], the *M. ilicifolia* plant drug must present at least 2.8 mg/g in the extract of epicatechin. All extracts prepared showed an epicatechin concentration higher than the preconized limit. The profile for the five extracts is presented in Figure 1. The CE1 presented the lower epicatechin concentration (mg/g of extract) of 3.6 ± 0.02, followed by the CE2 at 15.54 ± 0.14, CE4 at 16.77 ± 2.14, and CE3 at 19.63 ± 2.04. The CE5 presented a higher epicatechin concentration of 20.66 ± 1.99.

### 2.2. Antibacterial Activities of Extracts and Semi-Purified Fractions against H. pylori

Initially, the effect of the crude extracts, ethyl-acetate, *n*-butanolic, and aqueous fractions was investigated at different concentrations (32–1024 µg/mL), as shown in Table 1.

The best inhibitory activity corresponded to the CE5, EAF2, and EAF5 with an MIC_50_ of 64 µg/mL and *n*BF5 of 128 µg/mL. It was observed that partition with ethyl acetate demonstrated better MIC in comparison with other fractions.

The antibacterial result obtained is concordant with the result found in a previous study using *Plumbago zeylanica* L. The extract produced with acetone presented the lowest MICs when compared with the ethanol:water extract. In another study performed with *Sclerocarya birrea* (A.Rich.) Hochst, the result was similar, where the extract produced with acetone:water showed a MIC comparable with amoxicillin and metronidazole [20,21].

The fractions produced with the ethyl acetate and *n*-butanol for the ethanol:water 50:50 (*v*/*v*) and acetone:water 7:3 (*v*/*v*) extracts showed to be MIC between 64 and 256 µg/mL. This can be explained because when a crude extract is prepared using the mixture of acetone:water or when a fraction is obtained by the partition of crude extracts with ethyl acetate, the compounds that are mostly obtained in both cases are phenolic compounds [22].

The result obtained for the *n*BF inhibitory activity (128 µg/mL) can be compared with the result shown in the study using *Rosa hybrida* Colorado extract that is, it is rich in polyphenols compounds when the MIC was 10 µg/mL to the butanolic fraction and 100 µg/mL to the ethanolic fraction [23].

One of *H. pylori*’s virulence factors is the presence of the urease enzyme. This enzyme is responsible for allowing the bacteria to stay alive in the acid stomach environment [1,2]. The ability of *M. ilicifolia* extracts and fractions to inhibit the bacterial urease enzyme was evaluated. The results (Appendix A) showed that *n*BF5 presented the best inhibition (47.08%), followed by EAF2 (40.50%). *n*BF2 and EAF5 showed an enzyme inhibition of 37.27% and 37.51%, respectively.

The anti-*H. pylori* activity was tested for several traditional herbs in past studies and showed that *Cimicifuga heracleifolia* Kom could inhibit the urease enzyme by 42% [24]. It was related to the ability of several phenolic compounds to inhibit the urease enzyme. Compounds that had catechol as a skeleton showed the most potent inhibitory activities, ranging from 38 to 94%. This activity is attributed to the two ortho-hydroxyl groups that are presented in the aromatic ring of polyphenols molecules [25]. According to Pessuto [11], the number of polyphenolic hydroxyls present in the compound structure and the stereochemistry of the compounds are directly related to the ability to scavenge free radicals.

*M. ilicifolia* is a plant rich in polyphenols compounds, such as (epi)catechin (**I**), (epi)gallocatechin (**II**), procyanidins B1 and B2 (**III** and **IV**), quercetin-3-*O*-α-L-rhamnopyranosyl(1→6)-*O*-[*β*-D-glucopyranosyl(1→3)-*O*-α-L-rhamnopyranosyl(1→2)]-*O*-*β*-D-galactopyranoside (**V**), and kaempferol-3-*O*-α-L-rhamnopyranosyl(1→6)-*O*-[β-D-glucopyranosyl(1→3)-*O*-α-L-rhamnopyranosyl(1→2)]-O-β-Dgalactopyranoside (**VI**) maytefolins A and B (**VII** and **VIII**), (Figure 2). Due to the well-known presence of phenolic compounds in the specie, we can suggest that the anti-*H. pylori* activity observed is a result of their presence [11,26,27].

### 2.3. UPLC-MS Profiles of M. ilicifolia Extracts

Based on the anti-*H. pylori* assays, two fractions were chosen to follow the chemistry characterization. EAF2 and *n*BF5 were used in the UPLC-MS analysis.

Table 2 shows the retention time of the compounds separated for chromatography from EAF2, as well as the ion and correspondent fragment in the negative mode.

In the *n*BF5 fraction, it was characterized by glycosylated flavonoids such as kaempferol-galactoside-rhamnoside-rhamnoside and quercetin-rhamnopiranosyl-glucopiranoside-rhamonoside; besides this, the fraction presents some compounds that could not be identified (Appendix A).

Several studies have shown the presence of tannins and flavonoids in *M. ilicifolia* extracts that are considered responsible for their biological activity. Monomeric and dimeric flavonoids, such as epicatechin and procyanidins B1 and B2, were isolated and characterized in studies over the years from aqueous, hexanic, and acetonic extracts [11,28,29]. Glycosylated flavonoids are also present in the leaves extracted from *M. aquifolium* and *M. ilicifolia* and are composed of quercetin and kaempferol 3-*O*-glycosides [27,28,30,31]. Some authors have already demonstrated that procyanidins, catechin, and gallic acid isolated from natural products can have activity against *H. pylori* with reference strain and antibiotic-sensitive and resistant clinical isolates [32,33,34,35,36].

All these studies support the characterization proposed in the present work, considering that the same substances were isolated and characterized in those studies carried out with the specie. With the results obtained in the activity assays conducted with the fractions, we can presume that the phenolic compounds are responsible for the activity, as several authors demonstrated [11,27,28,30,31].

### 2.4. Antioxidant Capacity of Extracts and Semi-Purified Fractions

For the antioxidant capacity assays, seven extracts and fractions were selected that showed the best activity against *H. pylori*. The DPPH results are shown in Appendix A; the antioxidant capacity is present as IC_50_, which means that the extract concentration is needed to promote 50% of the antioxidant activity.

The antioxidant capacity for the tested extracts varies between 14.51 and 98.35 µg/mL. The more pronounced antioxidant capacity was observed in EAF4 with an IC_50_ 14.51 µg/mL followed by CE5 (IC_50_ 19.08 µg/mL) and EAF2 (IC_50_ 19.48 µg/mL), with no significant statistical difference. The worst antioxidant capacity in the DPPH assay was observed for *n*BF5 with an IC_50_ 98.35 µg/mL and the AQF3 (IC_50_ 94.72 µg/mL). The quercetin, used as a positive control, presented an IC_50_ of 2.99 µg/mL. It is known that the antioxidant capacity is more pronounced in the presence of polyphenols [11]. This can explain why the extract obtained with acetone and water was rich in phenolic compounds and had a more notable result.

A previous study tested the antioxidant capacity using the DPPH radical scavenging method of the ethyl-acetate and *n*-butanolic fraction from *M. royleana* (Wall. ex M.A. Lawson) Cufod, both rich in polyphenols. The ethyl-acetate fraction showed an IC_50_ of 55.01 µg/mL, while the *n*-butanolic fraction was 58.01 µg/mL. The antioxidant capacity was evaluated for the ethyl-acetate fraction from an acetonic crude extract by the DPPH radical scavenging method and showed a result of 25.39 µg/mL [11,37].

For the FRAP assay, the values of the antioxidant capacity were expressed as an mM Trolox/g extract equivalent (Appendix A). In this assay, the absorbance variance found was linearly proportional to the antioxidant concentration.

The FRAP assay showed results varying from 0.77 to 5.40 mM Trolox/g of the extract. The EAF2 and EAF4 showed the result of 5.40 and 5.15 mM Trolox/g of the extract, respectively, with no significant statistical difference. The quercetin was used as the positive control, and the antioxidant capacity was 15.41 mM Trolox/g of the extract.

Some species from the Celastraceae family were tested to determine the antioxidant capacity using the FRAP assay. They found *Cassine orientalis* (Jacq.) Kuntze e *M. pyria* (Willemet) N. Robson has an antioxidant capacity of 584 ± 5.24 e 190 ± 0.87 µM trolox/g in the extract [38].

Polyphenols have a great chemical structure for the removal of free radicals. For the best activity, these compounds must present some specific structural characteristics with the hydroxyl groups and in the ring substitution. Some structural specifications were tested previously, and it was confirmed that the presence of the ortho-hydroxyl group in the ring B increases the antioxidant capacity [39,40].

The compounds present in the extracts produced with organic solvents can eliminate free radicals more effectively than those compounds with more polarity present in aqueous extracts or fractions; the explanation could be the higher presence of phenolic compounds in the extracts produced using organic solvents.

The fact that FAE2 showed the best antioxidant capacity over the *n*BF could be explained by the fact that EAF2 is richer in phenolic compounds. These results are according to the conclusions of [11], who said that phenolic hydroxyl present in the phenolic compounds, as the fraction with the best antioxidant capacity and the one rich in phenolic compounds with orto-hydroxyl in the structure, are more capable of scavenging free radicals.

In summary, the present study demonstrated the anti-*H. pylori* activity of *M. ilicifolia* extracts and fractions represents a strong potential for use in the treatment or even more strongly acting in the prevention of *H. pylori* infection. The antioxidant capacity was confirmed for the specie using two different methods. Both activities can be attributed to the presence of phenolic compounds, such as monomeric, dimeric, and glycosylated flavonoids, which can be found in higher amounts in the extracts and fractions produced with organic solvents.

### 2.5. Virtual Screening

To identify the compound most likely to act as the *H. pylori* urease inhibitor from those present in the *M. ilicifolia* extract, four molecular docking simulations for each compound in the library using two different programs were carried out. In this way, the mean scores obtained from each program were used in the calculation of the mean relative score expressed by Equation (1). The compound kaempferol-3-galactoside-6-rhamnoside-3-rhamnoside, named CID 44258967, had the highest mean relative score, followed by the compound (epi)afzelechin-(epi)catechin-(epi)catechin, named AFZ-CAT-CAT (Figure 3). For these compounds, the best pose obtained in each docking simulation in the Gold program showed a recurrent conformation pattern in the urease active site (Appendix A), which suggests a site-specific interaction profile with the enzyme, which is characteristic of compounds with a drug-like behavior. Together, these data suggest that the compounds CID 44258967 and AFZ-CAT-CAT are the most likely urease inhibitors presented in the *M. ilicifolia* extract.

To describe the molecular interactions that occur between urease and the best MRS ligands, the PoseView program [41] was used. For the compound CID44258967 (Figure 4A), the program predicts the hydrogen bonds with residues His221, Glu222, Thr251, Ala278, and Arg338, hydrophobic contacts with the residues Met317, Leu318, Cys321, and Phe334, a π-π stacking with residue Phe334, and a charge-dipole interaction with one of the Ni^2+^ cofactors. For the compound AFZ-CAT-CAT (Figure 4B), hydrogen bonds are predicted with the residues Ala169, Glu222, Asp223, and Asp362, a hydrophobic contact with Met366 and charge-dipole interaction with the two Ni^2+^ cofactors.

There are no reports in the literature citing the activity of kaempferol-3-galactoside-6-rhamnoside-3-rhamnoside or (epi)afzelechin-(epi)catechin-(epi)catechin as urease inhibitors, which makes this work the first one to associate anti-ureolytic activity with these two compounds.

Regarding toxicity, as far as we know, there are no reports in the literature describing the possible toxic effects of these two compounds, which leads us to evaluate their toxicity in silico by the SwissADME server. As a result, the AFZ-CAT-CAT compound showed three violations of Lipinski’s rules, MW > 500, the number of N or O > 10, and the number of NH or OH > 5. In addition, it has an alert as a possible pan assay interference compound (PAIN). The compound CID44258967 presented the same three violations of Lipinski’s rules but no alert as PAIN. However, Lipinsky’s rules assess the usability of the compounds as oral drugs, not regarding their toxic effects. In this way, the extract, and fractions of *M. ilicifolia* used in this work, have already been evaluated for toxicity and showed no relevant effect on the mitochondrial activity of human stomach AGS cells (AGS, ATCC CRL-1739) [42].

## 3. Materials and Methods

### 3.1. Plant Material

The leaves of *M. ilicifolia* were collected in Marialva, Brazil (23°28′43″ S; 51°47′39″ W; 622 m altitude) on 16 February 2016. The plant material was identified and deposited in the State University of Maringa herbarium under registration number 29221. The material was collected with the permission of IBAMA-SISBIO and registered under the number 11995-3. Access to the botanical material was registered by the Sistema Nacional de Gestão do Patrimônio Genético e do Conhecimento Tradicional Associado, SisGen, under #AB65084.

### 3.2. Extracts and Fractions Preparation

The pulverized leaves were macerated with *n*-hexane (10%, *w*/*v*) for 10 days to decrease. Afterward, five extracts of 10% (*w*/*v*) were produced by turbo extraction (Ultra-turrax^®^—UTC115KT) using in *v*/*v* proportion:water (CE1); ethanol:water 50:50 (CE2); ethanol:water 70:30 (CE3); ethanol:water 96:4 (CE4), and acetone:water 7:3 (CE5). The solutions were filtered, evaporated on a rotary evaporator under reduced pressure at 40 °C, and lyophilized. The extracts were further partitioned in water with ethyl acetate and *n*-butanol. The fractios solutions were evaporated and lyophilized, resulting in ethyl-acetate (EAF), *n*-butanolic (*n*BF), and aqueous fractions (AQF) [11]. The crude extracts and fractions yield are presented in Appendix A.

### 3.3. Epicatechin Determination Using HPLC Method

The analyses were carried out using a Thermo^®^ HPLC (Thermo Electron, Waltham, MA, USA), with a PDA (photo diode array) spectrophotometry detector module (Model Finnigan™ Surveyor PDA Plus Detector-Thermo Electron, Waltham, MA, USA), integral pumps, and degasser (Finnigan™ Surveyor LC Pump Plus - Thermo Electron, Waltham, MA, USA) and autosampler (Finnigan™ Surveyor Autosampler Plus-Thermo Electron, Waltham, MA, USA) equipped with a 10 µL loop and controller software (Chromquest™, version 4.2, Thermo Electron, Waltham, MA, USA), a Phenomenex^®^ Gemini C-18 (250 × 4.6 mm, 5 µm) (Phenomenex^®^, Torrance, CA, USA), and a guard column (Phenomenex^®^ SecurityGuard™-RP C-18 cartridge) (Phenomenex^®^, Torrance, CA, USA). It was used as a mobile phase water:formic acid (pH 2.5) (phase A) and acetonitrile:formic acid (pH 2.5) (phase B). It used a linear gradient of 0 min 18% B; 13 min 25% B; 16 min 34% B; 20 min 42% B; 23 min 65% B; 25 min 18% B, with a flow of 0.8 mL/min, and 100 µL injection volume. Detection was performed at 210 nm [29].

The sample preparation was carried out following a previous study using 1 g of each crude extract instead the pulverized leaves. The standard solution’s preparation and obtention of the calibration curve were conducted according to the previous study (Epicatechin calibration curve: y = 151,061x + 2,726,000) [29].

### 3.4. Bacteria Strain

It used *H. pylori* strain ATCC^®^ 43504, amoxicillin sensitive, and metronidazole resistance. Bacteria were grown in the Columbia Agar supplemented with sheep blood (5%) in a 10.0% CO_2_ atmosphere at 37.0 °C for 72 h [43].

#### 3.4.1. Minimum Inhibitory Concentration (MIC) and Minimum Bactericidal Concentration (MBC)

The anti-*H. pylori* assay was conducted by the microdilution method [44]. As a positive control, amoxicillin and metronidazole (Sigma Chemical Co., St. Louis, MO, USA) were used. In each well, 100 µL of the sample solutions (32–1024 µg/mL) and 100 µL *H. pylori* suspension (≈10^6^–10^7^ bacteria/mL) were added both in supplemented BHI. The absorbance was measured at 620 nm and then incubated (37.0 °C/72 h/10.0% CO_2_). After incubation, the plate was homogenized, and a new measurement was performed to determine the MIC.

The MBC assay was performed for samples that presented MIC. The sample corresponding to the microplate well without apparent growth in BHI was harvested in a Columbia Agar plate (5% sheep blood) and incubated at 37.0 °C, 10.0% CO_2_, 72 h. The assay was determined by the lowest sample concentration able to inhibit colony formation.

#### 3.4.2. Urease Inhibition Assay

Urease inhibition activity was determined based on the production of ammonia catalyzed by the enzyme urease, according to the method described by [44]. The reaction microplate contained a mixture of 25 μL of urease 4 UI (Sigma Jack Bean urease type III) and 25 μL of the sample at varying concentrations, and it was incubated at room temperature for two hours. Then, 25 μL of phenol red (0.02%) and 200 μL of urea (50 mM) in 100 mM phosphate buffer (pH 6.8) were added to the microplate. After 20 min, the mixture absorbance was read at 540 mm using a microplate reader (iMark^®^, BioRad, Washington, DC, USA). Boric acid was used as the standard positive control for urease inhibition.

### 3.5. UHPLC-MS Conditions

UHPLC-MS analysis was performed on a Nexera X2 liquid chromatography system with an LC-30AD pump and Phenomenex^®^ Gemini C-18 column (250 mm × 4.6 mm) coupled with a Q-TOF Impact II (Bruker Daltonics, Bremen, Germany), with electrospray ionization source. The column was maintained at 40 °C with a linear elution gradient of water 0.1% formic acid (eluent A) and acetonitrile 0.1% formic acid (eluent B). The elution procedure follows 0–13 min 18% B; 13–16 min 25% B; 16–20 min 34% B; 20–23 min 42% B; 23–25 min 65% B; 25–28 min 18% B. The flow rate was 0.4 mL/min, and a 20 µL aliquot of each sample was injected.

The ESI was set in the Auto MS/MS acquisition mode, with an acquisition rate of 5 Hz (MS and MS/MS). The scans were acquired using the mass analyzer at 70–1500 *m*/*z*. The analyses were performed in a positive and negative ionization mode, with a capillary voltage of 4.00 kV, supply temperature of 220 °C, and desolvation gas flow of 8.0 L/min. Daughter-scan experiments were performed using the collision-induced dissociation (CID) obtained using a collision energy ramp in the range of 15–50 eV and collision gas pressure of 3.06 × 10^−3^ mBar in the collision chamber.

### 3.6. Antioxidant Capacity Assay

#### 3.6.1. Determination of DPPH (2,2-diphenyl-1-picrylhydrazyl) Radical Levels

*M. ilicifolia* extracts and fractions were diluted in methanol and prepared at final concentrations of 3.1–100.0 μg/mL. A total of 100 µL of freshly prepared 130 µM DPPH was added to 100 µL of the sample at different concentrations. The microplates were kept at room temperature and protected from light. After 30 min, the absorbances were measured at 517 nm in a microplate spectrophotometer (Biochrom Asys UVM 340-Cambridge, Cambridgeshire, England). Negative (methanol added to DPPH), white (methanol only), and positive (Trolox standard) controls were used [45].

#### 3.6.2. Ferric Reducing Antioxidant Power (FRAP) Assay

The FRAP assay method was described previously [45]. Trolox and samples (20–100 µg/mL) were diluted in ethanol. Absorbance values were measured using a microplate spectrophotometer (Biochrom Asys UVM 340 - Cambridge, Cambridgeshire, England) at 595 nm. To determine the total antioxidant capacity, it was used in the Trolox calibration curve equation (20–600 μM; y = 0.0028x + 0.0199 R² = 0.9993) from the sample’s absorbances. The results were expressed as Trolox equivalent antioxidant capacity (TEAC).

### 3.7. Virtual Screening

One of the beta subunits of the hetero 24-mer urease from *H. pylori* and bonded to the inhibitor 2-{[1-(3,5-dimethylphenyl)-1H-imidazol-2-yl]sulfanyl}-N-hydroxyacetamide (named DJM; CID 42065735) (PDB id: 6ZJA) was used in the virtual screening of compounds described in the *M. ilicifolia* extract (Appendix A). The programs and respective protocols were defined based on the redocking of the ligand in the urease. The AutoDock-Vina (Vina) program [46] used the standard search and ranking algorithms, with the search space defined by the box centered on the ligand, with dimensions of 30, 20, and 30 Å at x, y, and z, respectively. The uff force field [47] was used to minimize the ligands. The program Gold [48] uses the search space with a 15 Å radius centered on the ligand. The search algorithm employed a 200% efficiency, with 30 runs, and the ASP scoring and the ‘Allow early termination’ option was disabled.

The compounds described in the *M. ilicifolia* extract plus the reference ligand CID 42065735 were in the library for the virtual screening. The 2D or 3D structures of the compounds were obtained from the PubChem database or drawn by the ChemDraw^®^ program. Different isomers were used when necessary, and the OpenBabel program [49] was used to add hydrogens and convert them to 3D.

The mean relative score (*MRS*) was calculated from the mean docking score of each compound after four simulations using each program (Equation (1)). In this equation, *Vina* represents the average score of the compound obtained from four simulations using the *Vina* program, and *Vina_max_* expresses the maximum average score observed for the given compounds of the library; the same concept was applied to the results from the Gold program (*Gold* and *Gold_max_*). The *MRS* allows the ranking of compounds with the ligand CID42065735 as a reference.
(1)$$MRS=\frac{1}{2}\left(\frac{Vina}{Vina_{max}}+\frac{Gold}{Gold_{max}}\right)$$

The in-silico toxicity was estimated by the SwissADME server [50].

### 3.8. Statistical Analysis

Numerical data were presented as the means ± standard deviation (SD). The number of repetitions of individual assays was different and was present in the description of each of them. One-way ANOVA with Bonferroni post-test was performed considering values of *p* ≤ 0.05 as statistically significant.

## 4. Conclusions

The EAF2 and *n*BF5 presented the best in vitro anti-*H. pylori* activity among all the extracts and fractions tested. In the urease enzyme inhibition test, the highest percentage of inhibition was observed in the *n*BF5 and EAF2. The antioxidant capacity was better observed in the ethyl acetate fractions tested and for the CE5. The in silico assays demonstrated that kaempferol-3-galactoside-6-rhamnoside-3-rhamnoside and (epi)afzelechin-(epi)catechin-(epi)catechin presented in the fractions EAF2 and *n*BF5 and have an anti-ureolytic activity. The anti-*H. pylori* and antioxidant activity observed can be attributed to the phenolic compounds present in EAF2 and *n*BF5.

## Figures and Tables

**Figure 1 antibiotics-12-00046-f001:**
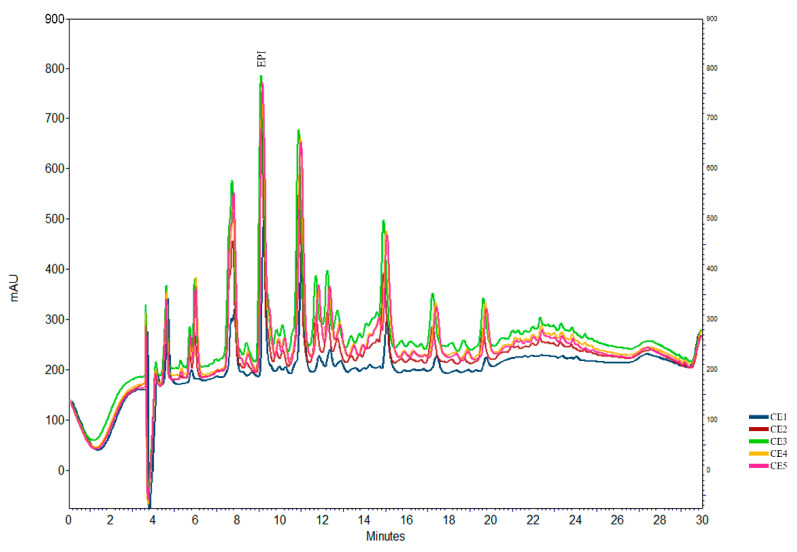
Chromatographic profile at 210 nm of the purified extract of *Monteverdia ilicifolia*. Chromatographic conditions: column Phenomenex^®^, Gemini C-18 (250 mm × 4.6 mm i.d., 5 μm), SecurityGuard (RP-cartridge) (20 mm × 4.6 mm i.d., 5 μm). Mobile phase: water (formic acid 0.1%) A and acetonitrile (formic acid 0.1%) B: 0 min 18% B; 13 min 25% B; 16 min 34% B; 20 min 42% B; 23 min 65% B; 25 min 18% B; flow-rate, 0.8 mL/min (CE1 = crude extract aqueous; CE2 = crude extract ethanol: water 50:50; CE3 = crude extract ethanol: water 70:30; CE4 = crude extract ethanol: water 96:4; CE5 = crude extract acetone: water 7:3).

**Figure 2 antibiotics-12-00046-f002:**
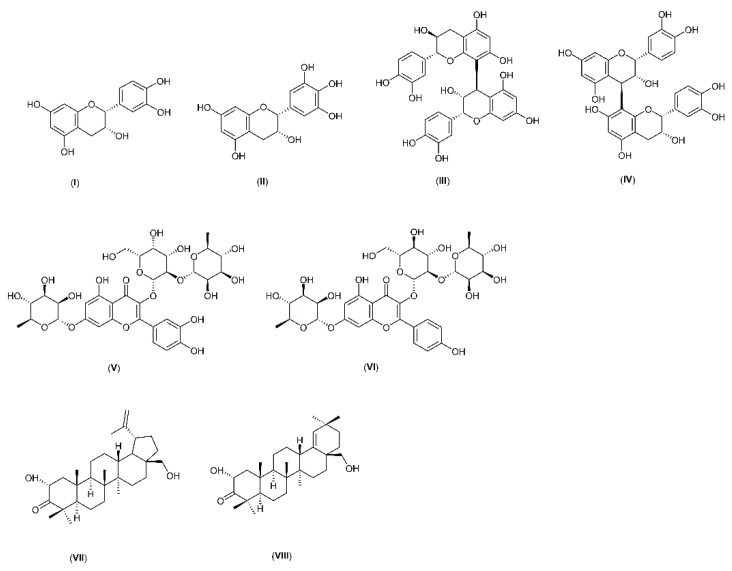
Chemical structures of compounds from *Monteverdia ilicifolia* (**I**–**VIII**) (Chem Draw v.14.0.0.118).

**Figure 3 antibiotics-12-00046-f003:**
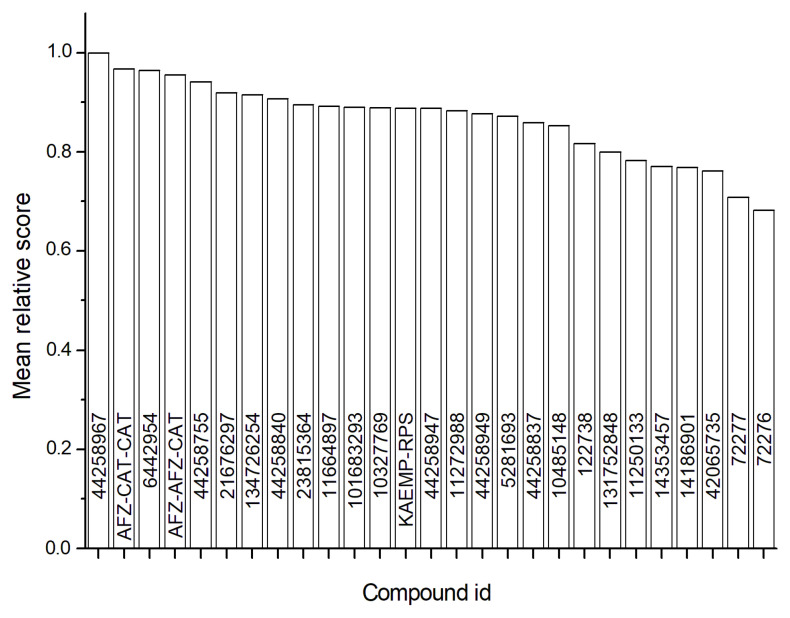
Mean relative docking scores of the compounds of the *Monteverdia ilicifolia* extract. The reference crystallographic ligand DJM (CID42065735) was used as reference.

**Figure 4 antibiotics-12-00046-f004:**
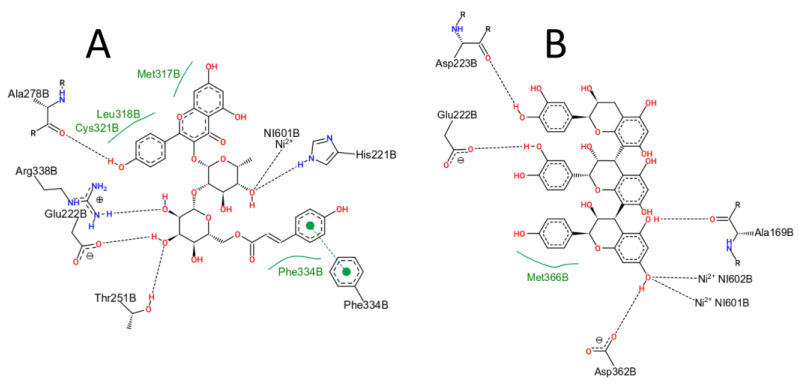
Intermolecular interactions between ligands Kaempferol-3-galactoside-6-rhamnoside-3-rhamnoside, CID44258967 (**A**) and (epi)afzelechin-(epi)catechin-(epi)catechin, AFZ-CAT-CAT (**B**), evaluated by the PoseView program.

**Table 1 antibiotics-12-00046-t001:** The MIC_50_ of *Monteverdia ilicifolia* extracts and fractions against *Helicobacter pylori*.

MIC_50_ (µg/mL ± SD)	CE1	CE2	CE3	CE4	CE5
CE	1024 ± 2.1	1024 ± 5.1	512 ± 5.6	512 ± 2.2	64 ± 9.8
EAF	>1024 ± 17.3	64 ± 5.4	256 ± 1.3	256 ± 3.5	64 ± 16.1
*n*BF	>1024 ± 4.7	256 ± 1.3	1024 ± 2.7	1024 ± 3.2	128 ± 4.0
AQF	>1024 ± 4.5	>1024 ± 0.7	512 ± 1.9	512 ± 5.1	256 ± 8.7

CE1 = crude extract aqueous; CE2 = crude extract ethanol: water 50:50; CE3 = crude extract ethanol: water 70:30; CE4 = crude extract ethanol: water 96:4; CE5 = crude extract acetone:water 7:3; EAF: ethyl-acetate fraction; *n*BF: *n-*butanolic fraction; AQF: aqueous fraction.

**Table 2 antibiotics-12-00046-t002:** Compound identification of EAF2 (ethanol: water 50:50 *v*/*v*) detected by UHPLC-HRMS negative mode.

Identification	Retention Time (Min)	[M-H]^-^ (*m*/*z*)	Main Fragments
(epi)gallocatechin	8.83	305	109, 125, 139, 165, 219, 237, 261
procyanidin B2	10.63	577	125, 151, 245, 289, 407, 425, 451
(epi)catechin	12.29	289	109, 179, 203, 245
(epi)afzelechin-(epi)catechin	13.70	561	125, 289, 435
(epi)afzelechin-(epi)catechin-(epi)catechin	14.77	849	125, 289, 407, 559, 679
(epi)afzelechin-(epi)afzelechin-(epi)catechin	15.31	833	125, 239, 407, 543
kaempferol-galactoside-rhamnoside-rhamnoside	15.51	739	284
(epi)catechin-(epi)catechin	17.17	577	125, 161, 289, 407
(epi)afzelechin-(epi)catechin	21.31	561	289, 435
kaempferol-rhamnopentoside	21.71	563	284

## Data Availability

Not applicable.

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
