# Peer review of "An In Vitro and In Silico Investigation about Monteverdia ilicifolia Activity against Helicobacter pylori"

_antibiotics, 2022, doi:10.3390/antibiotics12010046_

Round 1

Reviewer 1 Report

This is a manuscript about the bioactivities (inhibition against Helicobacter pylori, antioxidative activity) of extracts from Monteverdia ilicifolia, a Brazilian native plant. 

Commnets:

1) interrogative sentence is not good for the manuscript title.

2) some conclusions cannot be supported properly based on the results. Such as page 4, line 113-116, CE 2 is the extract by ethanol-water (50:50), no relationship with actone:water, EtOAc.

3) page 4, line 119-122, the citation about the extract of Hippocratea celastroides aganist H. pylori. I do not think it's a relevant citation.

4) in 2.2, some parts need be rewritten for the deficiency in logic.

Author Response

Dear Reviewer,

We would like to thank you for all contributions! We are certainly that all contributions will improve the manuscript’s quality. Below we detailed the responses of all requests.

Best regards,

Detailed response (authors response is written in italic letters):

1) interrogative sentence is not good for the manuscript title.

The manuscript title was rewritten.

2) some conclusions cannot be supported properly based on the results. Such as page 4, line 113-116, CE 2 is the extract by ethanol-water (50:50), no relationship with actone:water, EtOAc.

The conclusion expressed in the lines 113-116 was rewritten in order to express more clearly the original idea.

3) page 4, line 119-122, the citation about the extract of Hippocratea celastroides aganist H. pylori. I do not think it's a relevant citation.

We removed the reference. At first, we used this reference in order to compare our result with another vegetal specie tested against H. pylori.

4) in 2.2, some parts need be rewritten for the deficiency in logic.

The item 2.2 was rewritten in some parts and re-organised in order to establish a logical sequence.

Reviewer 2 Report

The current study presents some quite interesting results regarding the antimicrobial and antioxidant activity of Monteverdia ilicifolia extracts, as well as their chemical characterization. The methodology is solid, nevertheless there are some alterations that should be made before acceptance for publication:

- Remove the full stop mark “.” from the title; also in the title of the manuscript, either choose to start all words in capital letters or small letters, not a mixture of the two;

- In line 56, “Time and afford have been spent” should be “Time and effort have been spent”; please correct all mistakes like this throughout the manuscript, and improve the overall quality of the written English;

- “Materials and Methods” section should come before “Results and discussion” section; also there is a “Conclusion” section missing, that should be placed at the end of the manuscript;

- In line 82, you said you used the Brazilian Pharmacopeia, could you compare the methodology to the one described in international manuals, such as the United States Pharmacopeia or the European Pharmacopoeia?;

- Provide figure 1 in higher resolution;

- Provide a caption for figure 2, in page 5 (not correctly identified or described); then figure 2 should be renumbered as figure 3, and so on;

- Provide a graph of the results for section “2.4 Antioxidant capacity of extracts and semi-purified - fractions”.

Author Response

Dear Reviewer,

We would like to thank you for all contributions! We are certainly that all contributions will improve the manuscript’s quality. Below we detailed the responses of all requests.

Best regards,

Detailed response (authors response is written in italic letters):

- Remove the full stop mark “.” from the title; also in the title of the manuscript, either choose to start all words in capital letters or small letters, not a mixture of the two:

The final mark was removed. The capital letters used in the title are following the Guidelines for Authors that say: “Please capitalize all words in headings including hyphenated words (e.g. Anti-Antagonist), except conjunctions (and, or, but, nor, yet, so, for), articles (a, an, the), and all prepositions (including those of five letters or more) (in, to, of, at, by, up, for, off, on, against, between, among, under). First and last words in the title are always capitalized”.

- In line 56, “Time and afford have been spent” should be “Time and effort have been spent”; please correct all mistakes like this throughout the manuscript, and improve the overall quality of the written English:

The mistake pointed was correct and the manuscript were revised by an English native colleague.

- “Materials and Methods” section should come before “Results and discussion” section; also there is a “Conclusion” section missing, that should be placed at the end of the manuscript;

The sequence of sections was done according the Antibiotics Microsoft Word template file found at the Journal page, it is why the section “Material and Methods” is placed after the section “Results and Discussion”. As the section “Conclusion” is not mandatory we had left out, but now the section “Conclusion” was added in the manuscript.

- In line 82, you said you used the Brazilian Pharmacopeia, could you compare the methodology to the one described in international manuals, such as the United States Pharmacopeia or the European Pharmacopoeia?

We used the Brazilian Pharmacopoeia methodology because the species studied is a native's Brazilian flora, so is not presented in the American or European Pharmacopoeia. There isn’t any specie from the same genius that we could compare.

- Provide figure 1 in higher resolution;

The figure 1 was changed to a figure in better resolution.

- Provide a caption for figure 2, in page 5 (not correctly identified or described); then figure 2 should be renumbered as figure 3, and so on;

Thank you very much for your observation. The caption for figure 2 was added, and all figures were renumbered correctly in the captions and in the text.

- Provide a graph of the results for section “2.4 Antioxidant capacity of extracts and semi-purified - fractions”.

The graphs for the antioxidants results are presented in the supplementary material (Figure S1 and Figure S2).

Round 2

Reviewer 1 Report

The manuscript had been revised and was better than the first version. Alaos, there are some comments on the revised version.

1) in Figure 1, there are two images. They are nearly the same. Why do you give these two similar images. Should the chromatographic conditions be listed out in figure legend?

2) page 5, line 118, "a", not "an" can be used.

3) page 6, line 143, the number of polyphenolic hydroxyls is not related to the compounds sterochemistry. Please check it.

4) page 6, line 147, also in Figure 2, maytefolins A and B (I and II) are triterpenoids, not polyphenols. 

Author Response

Response to Review 1

Dear Reviewer,

Thank you very much for all comments that you made for us, that helped us to improve our manuscript.

All suggestions were considered and the answers are detailed below.

Best regards,

Detailed response (authors response is written in italic letters):

1) in Figure 1, there are two images. They are nearly the same. Why do you give these two similar images.

The first figure that appears was the old one in bad resolution, that was excluded, the second figure is the new one, with a better resolution. As we have to use the MS Word Track Changes both figures are appearing. The first figure will disappear after the changes were accepted.

Should the chromatographic conditions be listed out in figure legend?

Yes, when we present a chromatogram, it is standard that the parameters used in the run are presented in the legend of the chromatogram.

2) page 5, line 118, "a", not "an" can be used.

The correction was done: “when a crude extract is prepared”.

3) page 6, line 143, the number of polyphenolic hydroxyls is not related to the compounds sterochemistry. Please check it.

In this phrase the idea is that the number of polyphenolic hydroxyls and the compounds stereochemistry, separately, are related to the ability to scavenge free radicals.

The phase was rewrite: “the number of polyphenolic hydroxyls present in the compound structure, and the compounds stereochemistry are directly related to the ability to scavenge free radicals”.

4) page 6, line 147, also in Figure 2, maytefolins A and B (I and II) are triterpenoids, not polyphenols.

The text and the figure were corrected, showing the phenolic compounds and triterpenoids separately.
